# Stunting and Anemia in Children from Urban Poor Environments in 28 Low and Middle-income Countries: A Meta-analysis of Demographic and Health Survey Data

**DOI:** 10.3390/nu12113539

**Published:** 2020-11-18

**Authors:** Shireen Assaf, Christina Juan

**Affiliations:** The DHS Program, ICF, Rockville, MD 20850, USA; Christina.Juan@icf.com

**Keywords:** child malnutrition, stunting, anemia, urban poor, urbanicity, urbanization, Demographic and Health Surveys (DHS), meta-analysis, urban–rural residence

## Abstract

Child malnutrition remains a global concern with implications not only for children’s health and cognitive function, but also for countries’ economic growth. Recent reports suggest that global nutrition targets will not be met by 2025. Large gaps are evident between and within countries. One of the largest disparities in child malnutrition within counties is between urban and rural children. Large disparities also exist in urban areas that have higher rates of child malnutrition in the urban poor areas or slums. This paper examines stunting and anemia related to an urban poverty measure in children under age 5 in 28 low and middle-income countries with Demographic and Health Survey data. We used the United Nations Human Settlements Programme (UN-HABITAT) definition to define urban poor areas as a proxy for slums. The results show that in several countries, children had a higher risk of stunting and anemia in urban poor areas compared to children in urban non-poor areas. In some countries, this risk was similar to the risk between the rural and urban non-poor. Tests of heterogeneity showed that these results were not homogeneous across countries. These results help to identify areas of greater disadvantage and the required interventions for stunting and anemia.

## 1. Introduction

Child stunting and malnutrition remain a global concern. Globally, among children under age five, at least one in three are not growing as they should because of malnutrition that includes stunting. At least one in two children under age five also suffer from deficiencies in vitamins and essential nutrients [1]. Despite declines in stunting among children globally, progress has been too slow to achieve the 2025 global nutrition target of reducing stunting by 40% to a target of 100 million; it is projected that 130 million children will be stunted by 2025 [2]. There are an estimated 149 million (or almost a quarter) of children under age five who are stunted [2]. There have also been global reductions in anemia prevalence over the last several decades; although, the total cases of anemia increased from 1.42 billion in 1990 to 1.74 billion in 2019 [3]. The greatest burden of anemia is found in western and Central Sub-Saharan Africa and South Asia [3,4], with the highest combined prevalence of anemia (39.7%) found in children under age five [3]. Malnutrition in children not only has a direct effect on their growth, but also on their cognitive, motor, and language development, mental health, susceptibility to diseases, and other long-term development such as lower levels of learning capacity that can continue into adulthood [2,5,6,7,8]. Malnutrition also influences the economic development and growth of countries and contributes to the intergenerational transmission of poverty [2]. 

Although there are disproportionately higher rates of child malnutrition in parts of Asia and Africa, within-country disparities can also be substantial in a growing population with rapid urbanization. As cities across the world continue to grow, the increase in urbanization can be coupled with an increase in urban poverty [9,10]. The pace at which the world’s population grows is notable especially in low and middle-income countries (LMICs). More than two-thirds (68%) of the world’s population is expected to live in urban areas by 2050 [11]. The process of urbanization, especially in LMICs, can lead to negative health outcomes overall and specifically for children. In particular, large disparities in child malnutrition exist between urban and rural areas, geographical regions, and levels of household wealth [1,2,12,13,14,15,16,17]. In one study that analyzed household survey data from 47 countries, rural/urban differences in stunting decreased but persisted and remained significant after controlling for household wealth [18]. The same study found that urban poor can be just as disadvantaged as the rural poor in terms of stunting and under-five mortality [18]. Other studies that examined urban poor areas or slums found higher rates of child malnutrition and mortality in these areas compared to the urban non-poor areas or non-slums [19,20,21].

In addition to the burden of undernutrition, countries increasingly face the double burden of undernutrition and overnutrition (i.e., overweight and obesity) [22,23]. This double burden of malnutrition can be found at the individual, household, and population levels [23]. A child can be overweight and also have poor nutrition. In addition, children who are stunted in childhood have a higher risk of becoming overweight in adulthood when they consume energy-dense diets and live a sedentary lifestyle [23,24]. The LMICs also face this double burden at the population level where there are high rates of undernutrition and overweight or obesity [23,25,26,27]. There are several causes to this phenomenon including the global nutrition transition caused by shifts in the agricultural system and the growth of nutrient-poor foods that are high in sugar, unhealthy fats, and/or are highly processed [26,28]. Such changes have especially affected urban areas and particularly the urban poor because healthier, more nutritious food is costly and difficult to access [26,28,29]. Physical activity has also declined, especially in urban areas, which further aggravates the problem. Countries need an integrated intervention approach that addresses undernutrition (wasting, stunting, and micronutrient deficiency) and the increasing burden of overweight, obesity, and diet-related non-communicable diseases [23,26,29]. 

It is important to study within-country disparities in child malnutrition to identify areas where interventions and programs should be targeted, and to go beyond the urban–rural dichotomy. In this paper, we focus on the urban poor areas as a proxy for slum areas and if children in these areas experience higher rates of stunting and anemia compared to children in the urban non-poor areas. We also explore if differences in child malnutrition between the urban poor and urban non-poor are similar to or different from differences between the rural and urban non-poor areas. Given the rapid increases in urbanization and urban poverty, it is important to continually quantify and examine disparities in health outcomes within urban areas. 

## 2. Data and Methods

### 2.1. Data

Data from countries that had a Demographic and Health Survey (DHS) from 2014 to 2018 were included in the analysis. Countries with a population less than 10 million were excluded, which left 30 surveys that were considered for the analysis. These surveys use a randomized, multi-stage sample design that produces estimates that are representative of the country at the time of the survey. The descriptive analysis revealed that Jordan and Zimbabwe did not have any children in urban poor clusters according to the definition used in this paper. Therefore, these two countries appear in the results for the description of the urban poor cluster variable in Figure 1 only. Table 1 shows the countries in the analysis and their sample sizes. 

### 2.2. Variables

The outcomes of interest are stunting, and moderate to severe anemia in children. Stunting is defined as the proportion of de facto children under age five who have a height-for-age z-score below the median of the WHO 2007 reference population by more than two standard deviations. Moderate to severe anemia in children is defined as the proportion of de facto children age 6–59 months with a hemoglobin level less than 10 g per deciliter. Hemoglobin levels are adjusted for altitude.

The main variable of interest is the urban poverty cluster variable. To construct this variable, we used the UN-HABITAT definition of a slum household, which is a household that lacks one or more of the following: durable housing of permanent nature, sufficient living space for not more than three persons per room, access to safe water, access to adequate sanitation, and security of tenure that prevents forced evictions [31,32]. The DHS data includes information on all items except the security of tenure. Therefore, we defined an urban poor household as households in an urban cluster that are lacking two or more of the following: durable material for the floor, wall, and roof; access to improved water; access to improved sanitation; and fewer than three persons per sleeping room. To define the variable at the cluster level, an urban poor cluster was classified as a cluster with more than half of the households classified as urban poor households, as recommended by United Nations Human Settlements Programme [9]. Clusters that contained fewer than half urban poor households were defined as urban non-poor areas. This aggregation to the cluster level was used to define areas that may be considered slum-like (the urban poor). Therefore, the urban poverty cluster variable has three categories: rural, urban poor, and urban non-poor.

The control variables used in the regressions include the child’s sex, age in months categorized in 6-month intervals, the mother’s education level, and the region at the administrative level 1 for the country. The wealth index could not be included as a control because it was highly correlated with the urban poverty cluster variable. 

### 2.3. Methods

The analysis includes descriptive and logistic regression analysis. A crosstabulation of the urban poverty cluster variable and each outcome was performed along with Chi-square tests of association. In addition, adjusted logistic regressions were fit for each outcome and the variable of interest along with the controls. The results were summarized with forest plots for the main variable of interest that also produce a pooled odds ratio using meta-analysis and the "metan" command in Stata [33]. The metan command also performs a heterogeneity test that examines the variability between the surveys. This test produces an I-squared index and *p*-value. An I-squared value of 25–50% is considered a low level of heterogeneity, 50–75% is moderate, and greater than 75% is high [34]. A significant *p*-value would indicate that we reject the null hypothesis that the surveys are homogeneous for the result and that the surveys are heterogeneous. For the meta-analysis, the surveys were assigned equal weight so that large surveys such as India did not predominate the results. 

All analyses accounted for the sampling design and sampling weights and were performed with Stata 16 software. The estimates are reported with 95% confidence intervals (CI).

## 3. Results

Figure 1 shows the percentage distribution of the population of urban non-poor, urban poor, and rural clusters for the countries in the analysis. Several countries had a highly rural population with more than three-quarters of the population in Burundi, Cambodia, Chad, Ethiopia, Malawi, Mali, Rwanda, and Uganda living in rural clusters. The highest proportion of people who live in urban poor clusters were found in Kenya and Benin (both 17%), followed by the Democratic Republic of the Congo (DRC) and Chad (both 16%). Jordan was highly urban with no urban poor clusters. In contrast, Zimbabwe was highly rural with no urban poor clusters. Therefore, the remaining analyses do not include Jordan and Zimbabwe. Nineteen of the 30 countries shown in Figure 1 have fewer than 5% of the population living in urban poor clusters. 

**Figure 1 nutrients-12-03539-f001:**
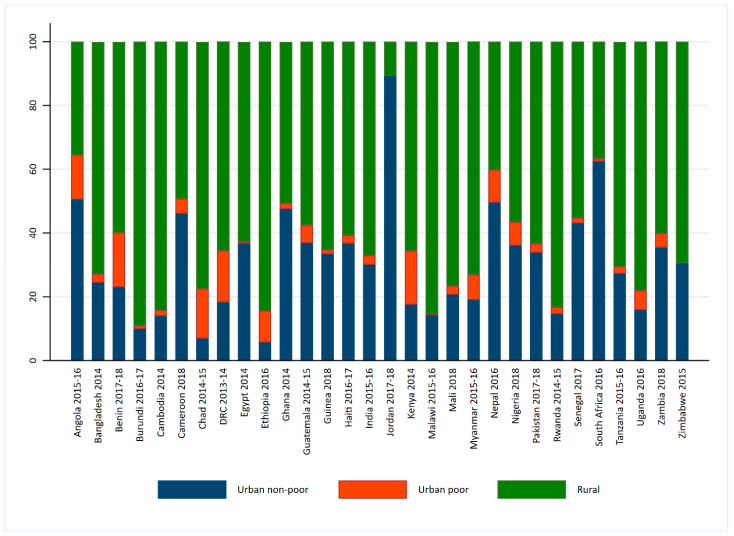
Distribution of the urban poverty cluster variables for the countries in the analysis.

Table 2 shows the percentage of children under age five who are stunted. More than one-third of children under age five were stunted in 15 countries during the time of the survey: Angola, Bangladesh, Burundi, Chad, DRC, Ethiopia, Guatemala, India, Malawi, Nepal, Nigeria, Pakistan, Rwanda, Tanzania, and Zambia. Generally, the lowest percentage of stunting was found in urban non-poor clusters and the highest in rural clusters. However, in Bangladesh, Guatemala, Haiti, India, Nepal, Pakistan, Senegal, Tanzania, and Uganda, stunting was highest in urban poor clusters with the highest found in Pakistan (56%), Guatemala (55%), Bangladesh (48%), and Burundi (45%). Differences in stunting across the urban poverty cluster distribution was significant except for Egypt and South Africa. 

Since four surveys in Table 3 did not perform anemia testing in children, the analysis with the anemia outcome was performed on 24 surveys. The highest level of moderate to severe anemia in children age 6–59 months was found in Mali (57%), followed by Guinea and Benin, both at 43%. Over one-third of children from 12 of the 24 countries had moderate to severe anemia. In general, anemia was lowest in children who live in urban non-poor clusters and higher in the rural areas. However, in several surveys, there was either no significant difference in anemia between the urban non-poor and rural clusters or anemia was the highest in the urban poor clusters. More than 40% of children age 6–59 months who live in urban poor clusters in Benin, Ghana, Haiti, Mali, Nigeria, and Senegal had moderate to severe anemia. In Angola, Myanmar, South Africa, Uganda, and Zambia, there were no significant differences in moderate to severe anemia across the urban poverty clusters. 

Figure 2 and Figure 3 summarize the regression results for the stunting and anemia outcomes and the urban poverty cluster variable. These estimates can also be found in Appendix A and Appendix B. The figures show the adjusted odds ratios for the rural and urban poor categories compared to the urban non-poor reference category. In Figure 2, there are more significant findings for rural compared to urban non-poor than urban poor compared to urban non-poor. With Egypt as an exception, children in rural clusters are more likely to be stunted than children in the urban non-poor clusters. The pooled odds ratio from the meta-analysis was 1.5 (95% CI 1.5, 1.6). Overall, living in a rural area increases the risk of stunting in children by 50% compared to living in an urban non-poor cluster. However, the I-squared results indicate that the surveys were heterogeneous. Several countries (16 from the 28) exhibited significantly higher odds of stunting in children who live in urban poor clusters compared to urban non-poor with the adjusted odds ratios ranging from 1.3 to 2.0, except for Egypt and Ghana. In Ethiopia, Guatemala, Pakistan, and Tanzania, children living in urban poor clusters had almost twice the odds of being stunted compared to children in the urban non-poor clusters. For India, Mali, and Nigeria, the odds ratios for both the rural and urban poor cluster categories were very similar, which indicated that children living in these areas had a similar risk of stunting in comparison to children living in urban non-poor clusters. The pooled odds ratio from the meta-analysis was 1.3 (95% CI 1.2, 1.5), which indicated that children living in urban poor clusters have a 30% increased odds of stunting compared to children living in urban non-poor clusters. The I-squared results also indicate that the surveys were heterogeneous. In Egypt and Ghana, children in urban poor clusters had almost 50% lower odds of being stunting compared to children in the urban non-poor clusters. In Egypt, children in rural clusters also had a 50% lower odds of being stunting compared to children in urban non-poor clusters.

As shown in Figure 3, in 13 of 24 surveys, children living in rural clusters had a higher odds of having moderate to severe anemia compared to children living in urban non-poor clusters. In 12 of 24 surveys, children living in urban poor clusters had greater odds of having moderate to severe anemia compared to children living in urban non-poor clusters. For several countries, the odds of children living in urban poor clusters having anemia compared to children in urban non-poor clusters were almost twice as high. For Burundi, Nigeria, and Senegal, the odds ratios for the rural and urban poor categories were very similar, which indicated that children living in these clusters had a similar risk of anemia when compared to children living in urban non-poor clusters. The pooled odds ratio from the meta-analysis for children living in rural and urban poor clusters compared to urban non-poor clusters were both 1.4. This means that children in rural and urban poor clusters had 40% greater odds of being anemic compared to children who live in the urban non-poor clusters. However, the I-squared index and *p*-value indicate that this result has moderate to high level of heterogeneity. One unexpected finding was the 60% lower odds of moderate to severe anemia in children living in urban poor clusters compared to the urban non-poor children in Guinea.

## 4. Discussion

This paper examined the prevalence of stunting and moderate to severe levels of anemia among children under age five in 28 countries. Globally, the world’s population continues to grow, and urbanization continues to increase with 84% of the world’s population living in less developed regions [11]. It is important to examine disparities within countries that account for the nuanced needs of people based on where they live. This study examined stunting and anemia through an urbanicity measure [21,35], referred to here as the urban poverty cluster variable, which goes beyond the traditional urban–rural residence dichotomy used in other studies. With this multi-country DHS meta-analysis that used the urban poverty cluster measure, we were able to focus on areas of greater disadvantage based on these two malnutrition conditions in children that have long-term developmental consequences for the child, as well as consequences for the society as a whole [2,5,6,7,8,22].

In more than half the countries in our analysis, children in urban poor clusters had a greater odds of being stunted than those in urban non-poor clusters. Similarly, in half of the countries in our analysis with available data on anemia, children in urban poor clusters had a higher odds of having moderate to severe anemia than children in urban non-poor clusters. In an 11-country DHS analysis focused on intra-urban differentials, researchers found that child stunting was as much a problem in urban areas as in rural areas when socioeconomic status (SES) was considered within these areas [36]. In this same study, children in Guatemala were seven times more likely to be stunted in urban low SES areas, compared to the urban high SES areas. Similar relationships were found in Pakistan and Tanzania, but at lower magnitudes. These three countries were included in our study, and displayed greater odds of stunting among children in urban poor clusters compared to urban non-poor clusters. Previous work that used data from 73 countries with DHS data found that children in slums have worse health risks compared with children in other urban areas, but lower risks than their rural counterparts [20]. The authors concluded that a large share of these differences can be explained by differences in the mother’s education, wealth, and access to health services. In another study that examined health outcomes in Bangladesh, Kenya, Egypt, and India, children living in urban-slums fared worse in terms of health outcomes, including anemia and stunting, compared to their urban non-slum and rural counterparts [37].

In addition to expected findings, there were also some noteworthy, unexpected findings. We found that children in urban poor clusters had lower odds of being stunted in two countries—Egypt and Ghana—compared to children in urban non-poor clusters. Rural children in Egypt also displayed lower odds of being stunted compared to children in urban non-poor clusters. This aligns with the finding in another study in Egypt where stunting is decreasing among the poorest wealth quintiles but increasing in the richest wealth quintiles [38]. In addition, food programs in Egypt have focused on energy-dense foods, especially among poor and rural populations, given that rising food prices and food insecurity have led to an increase in consumption of calorie-dense foods [38,39]. Egypt and Ghana are also experiencing the double burden of malnutrition [39,40], and in Ghana, concurrent overweight and stunting was found to be significantly higher in children from the fourth wealth quintile compared to children from the poorest wealth quintile [40]. More research is needed to understand the changes in nutrition transition and dietary diversity in these countries that may shed more light on these findings. Another area for further exploration is the high provision of services in these countries’ urban poor areas, that may counter the negative effects of living in these slum-like conditions [21,37,41,42]. For example, one study found that living in a slum neighborhood was associated with poor child outcomes, except when mothers had access to and received health services (such as antenatal care) from trained providers [21]. Further research is required to explore these country-specific findings. The relationship between urbanization and child malnutrition is complex and may be modified by some advantages and disadvantages to living in an urban environment [14,29,43].

Our study has some limitations. The urban–rural variable was used when constructing the urban poverty cluster variable, our main variable of interest. Since the urban–rural dichotomous variable available in the DHS data uses each country’s specific definition of what is urban or rural, this definition is country-specific and not standardized [44]. Incorporating this DHS urban–rural residence variable has the potential to introduce misclassification and error into the urban poverty cluster variable used in our analysis. Another potential limitation of our study is that the UN-HABITAT’s definition of a slum household [31], which we used in the construction of the urban poverty cluster variable, may be more pertinent to lower-income countries than to the middle-income countries. There is potential for missed opportunities in capturing within-country inequalities because some domains in the UN-HABITAT’s definition may not be as relevant. Some countries’ circumstances require other domains to be incorporated for better comparability in the definition of a slum household. Finally, the displacement procedures employed by the DHS mask the real locations of urban clusters by a displacement of up to 2 km. Therefore, it is not possible to know if the urban poor cluster we identify is actually a slum area or another disadvantaged area. Thus, we exercised caution when identifying the urban poor clusters in our analysis as actual slums.

Based on this study, we recommend that researchers, policymakers, and program implementers go beyond the usual urban–rural measurement to examine poor child health outcomes, as recommended by previous researchers [20,36,45]. We suggest considering the inclusion and use of an urban poverty cluster variable based on available data per domain to provide a better understanding of urban and rural environments. Country-specific programming and policies focused on the rural and urban poor clusters are essential. Given the similarities in the odds of stunting and moderate to severe levels of anemia across these two cluster categories in some countries, programming might consider accounting for these findings in complementary ways.

## 5. Conclusions

This study further informs the relationship between more nuanced areas of urban and rural environments and children’s risk of being stunted and experiencing moderate and severe levels of anemia. In several countries, we found that children living in urban poor areas experienced worse child malnutrition outcomes compared to those in the urban non-poor areas. In some countries in our analysis, this disparity was similar to the disparity between the rural and urban non-poor. Future research with the urban poor cluster variable should consider contextual factors in urban poor clusters and slum areas, such as the availability and use of health services and accessibility to nutrient-rich foods. Intervention programs should consider urban areas that may be further disadvantaged if they are identified as slums or areas with poor access to water, sanitation, and other issues such as overcrowding, which can have direct consequences on children’s health outcomes including malnutrition.

## Figures and Tables

**Figure 2 nutrients-12-03539-f002:**
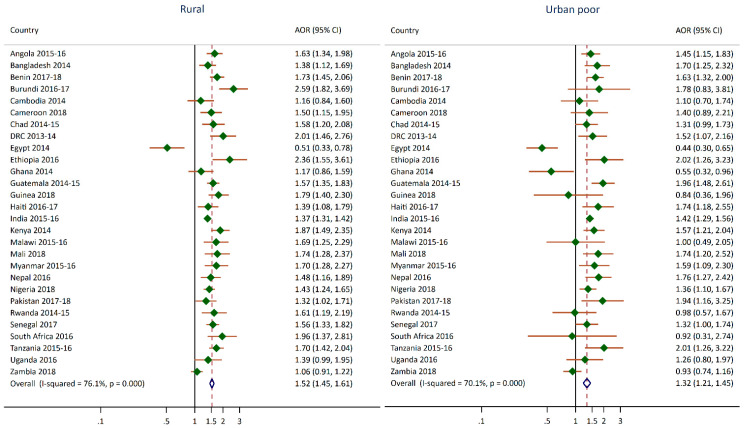
Adjusted odds ratios of stunting of children living in urban poor and rural clusters compared to children living in urban non-poor clusters.

**Figure 3 nutrients-12-03539-f003:**
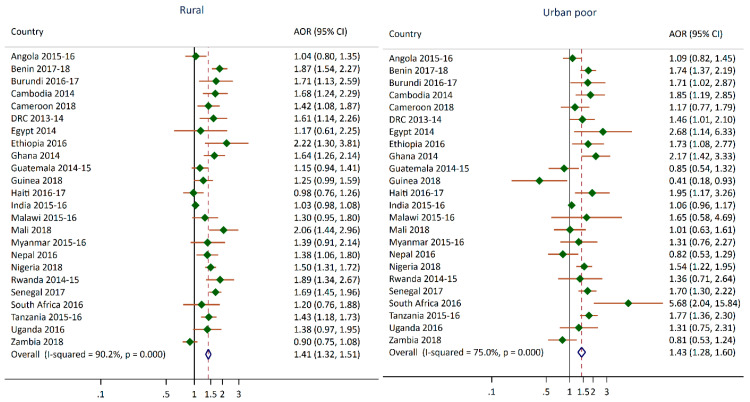
Adjusted odds ratios of moderate or severe anemia for children living in urban poor and rural clusters compared to children living in urban non-poor clusters.

**Table 1 nutrients-12-03539-t001:** Surveys used in the analysis with sample sizes and population size.

Country	DHS Survey	Number of Households Interviewed	Projected Population in 2020 (Thousands) *
Angola	2015–2016	16,109	32,866
Bangladesh	2014	17,300	164,689
Benin	2017–2018	14,156	12,123
Burundi	2016–2017	15,977	11,891
Cameroon	2018	11,710	26,546
Cambodia	2014	15,825	16,719
Chad	2014–2015	17,233	16,426
DRC	2013–2014	18,171	89,561
Egypt	2014	28,175	102,334
Ethiopia	2016	16,650	114,964
Ghana	2014	11,835	31,073
Guatemala	2014–2015	21,383	17,916
Guinea	2018	7912	13,133
Haiti	2016–2017	13,405	11,403
India	2015–2016	601,509	1,380,004
Jordan	2017–2018	18,802	10,203
Kenya	2014	36,430	53,771
Malawi	2015–2016	26,361	19,130
Mali	2018	9510	20,251
Myanmar	2015–2016	12,500	54,410
Nepal	2016	11,040	29,137
Nigeria	2018	40,427	206,140
Pakistan	2017–2018	11,869	220,892
Philippines	2017	27,496	109,581
Rwanda	2014–2015	12,699	12,952
Senegal	2017	8380	16,744
South Africa	2016	11,083	59,309
Tanzania	2015–2016	12,563	59,734
Uganda	2016	19,588	45,741
Zambia	2018	12,831	18,384
Zimbabwe	2015	10,534	14,863

Note: * Figures extracted from [30]. DHS, Demographic and Health Surveys; DRC, Democratic Republic of the Congo.

**Table 2 nutrients-12-03539-t002:** Percentage and 95% CI of children under age 5 that are stunted with crosstabulation by urban poverty cluster variable.

Survey	Total	Urban non-Poor	Urban Poor	Rural	*p*-Value
Angola 2015–2016	37.6 (35.7,39.5)	28.2 (24.5,32.3)	41.4 (37.8,45.0)	45.7 (43.5,47.9)	0.001
Bangladesh 2014	36.1 (34.4,37.9)	28.5 (25.0,32.2)	47.6 (40.1,55.2)	37.9 (35.9,39.9)	0.001
Benin 2017–2018	32.2 (30.9,33.4)	21.7 (19.6,23.9)	33.9 (31.2,36.8)	35.2 (33.7,36.8)	0.001
Burundi 2016–2017	55.9 (54.2,57.7)	25.7 (20.1,32.2)	45.1 (24.8,67.1)	58.8 (57.0,60.5)	0.001
Cameroon 2018	28.9 (27.1,30.8)	18.1 (15.7,20.8)	32.7 (23.0,44.2)	36.2 (33.7,38.8)	0.001
Cambodia 2014	32.4 (30.6,34.3)	22.5 (18.9,26.6)	30.9 (24.2,38.5)	33.8 (31.8,35.9)	0.001
Chad 2014–2015	39.9 (38.4,41.3)	25.0 (21.3,29.1)	35.0 (31.7,38.5)	41.6 (39.9,43.4)	0.001
Democratic Republic of the Congo 2013–2014	42.7 (40.9,44.5)	25.1 (20.5,30.3)	39.0 (35.3,42.9)	47.1 (44.9,49.4)	0.001
Egypt 2014	21.4 (20.1,22.9)	23.1 (20.5,26.0)	15.9 (12.3,20.4)	20.7 (19.1,22.4)	0.112
Ethiopia 2016	38.4 (36.5,40.3)	14.6 (11.5,18.4)	29.8 (23.6,36.7)	39.9 (37.9,42.0)	0.001
Ghana 2014	18.8 (17.0,20.6)	14.8 (12.4,17.6)	15.1 (11.4,19.8)	22.1 (19.7,24.7)	0.001
Guatemala 2014–2015	46.5 (44.8,48.2)	30.0 (27.8,32.3)	55.1 (47.5,62.4)	53.0 (50.8,55.1)	0.001
Guinea 2018	30.3 (28.6,32.1)	21.7 (18.9,24.9)	21.3 (10.5,38.5)	33.8 (31.8,35.9)	0.001
Haiti 2016–2017	21.9 (20.5,23.5)	16.8 (14.6,19.2)	29.4 (22.0,38.1)	23.9 (22.0,25.9)	0.001
India 2015–2016	38.4 (38.1,38.7)	29.4 (28.6,30.2)	42.6 (40.8,44.5)	41.2 (40.8,41.5)	0.001
Kenya 2014	26.0 (25.1,27.0)	16.3 (13.6,19.4)	23.2 (21.1,25.4)	29.1 (27.9,30.2)	0.001
Malawi 2015–2016	37.1 (35.6,38.7)	25.0 (20.7,29.8)	(24.3) (16.0,35.2)	38.9 (37.2,40.6)	0.001
Mali 2018	26.9 (25.6,28.2)	15.4 (13.6,17.5)	27.8 (21.2,35.5)	29.4 (27.9,30.9)	0.001
Myanmar 2015–2016	29.2 (27.3,31.1)	17.0 (13.9,20.6)	25.1 (19.8,31.3)	31.6 (29.5,33.9)	0.001
Nepal 2016	35.8 (33.5,38.3)	28.3 (25.0,31.7)	44.1 (37.6,50.8)	40.2 (36.6,43.9)	0.001
Nigeria 2018	36.8 (35.6,38.1)	24.2 (22.3,26.3)	39.6 (35.7,43.6)	44.8 (43.2,46.3)	0.001
Pakistan 2017–2018	37.6 (34.8,40.6)	28.4 (24.9,32.2)	55.7 (44.5,66.4)	40.9 (37.1,44.9)	0.001
Rwanda 2014–2015	37.9 (36.1,39.6)	22.7 (19.0,26.9)	28.9 (15.8,47.0)	40.6 (38.6,42.6)	0.001
Senegal 2017	16.5 (15.6,17.5)	9.5 (8.3,10.8)	23.3 (18.6,28.8)	20.2 (19.0,21.4)	0.001
South Africa 2016	27.4 (24.3,30.7)	26.0 (20.9,31.7)	ND	29.2 (25.8,32.8)	0.338
Tanzania 2015–2016	34.4 (33.0,35.9)	22.8 (20.5,25.3)	39.5 (24.1,57.3)	37.8 (36.1,39.4)	0.001
Uganda 2016	28.9 (27.3,30.5)	20.0 (16.5,24.1)	31.6 (25.6,38.2)	30.2 (28.4,32.0)	0.001
Zambia 2018	34.6 (33.4,35.8)	31.9 (29.6,34.4)	33.8 (28.9,39.1)	35.9 (34.4,37.3)	0.016

Note: The *p*-value is produced from the Chi-square test of the association between the urban poverty cluster variable and stunting. ND is not displaced because the estimate is based on fewer than 25 observations. Estimates in parentheses are based on 25–50 observations.

**Table 3 nutrients-12-03539-t003:** Percentage and 95% CI of children age 6–59 months with moderate or severe anemia with crosstabulation by urban poverty cluster variable.

Survey	Total	Urban non-Poor	Urban Poor	Rural	*p*-Value
Angola 2015–2016	34.1 (32.2,36.1)	32.6 (29.7,35.7)	35.3 (31.1,39.7)	35.1 (32.0,38.3)	0.419
Bangladesh 2014	NA	NA	NA	NA	
Benin 2017–2018	43.9 (42.2,45.6)	28.9 (26.1,31.9)	47.5 (43.5,51.6)	47.8 (45.6,50.0)	0.001
Burundi 2016–2017	36.3 (34.6,38.1)	23.6 (17.8,30.7)	31.7 (26.2,37.9)	37.5 (35.6,39.3)	0.001
Cameroon 2018	31.0 (29.1,33.0)	25.5 (22.8,28.5)	30.7 (23.9,38.5)	34.8 (32.0,37.8)	0.001
Cambodia 2014	25.7 (24.0,27.5)	15.7 (12.6,19.3)	29.2 (24.2,34.9)	27.0 (25.0,29.0)	0.001
Chad 2014–2015	NA	NA	NA	NA	
Democratic Republic of the Congo 2013–2014	34.8 (32.5,37.1)	26.6 (23.3,30.1)	35.5 (30.3,41.1)	36.2 (33.1,39.3)	0.005
Egypt 2014	9.5 (8.3,10.7)	6.2 (4.8,7.9)	(14.0) (4.0,38.7)	11.0 (9.5,12.7)	0.001
Ethiopia 2016	32.0 (29.5,34.6)	21.7 (16.1,28.5)	26.2 (20.8,32.4)	32.8 (30.0,35.6)	0.011
Ghana 2014	39.1 (36.3,41.9)	29.6 (25.9,33.6)	52.2 (37.5,66.5)	46.2 (42.6,49.8)	0.001
Guatemala 2014–2015	12.1 (11.3,13.0)	9.3 (8.1,10.6)	9.1 (6.8,12.1)	13.6 (12.5,14.8)	0.001
Guinea 2018	43.8 (41.6,46.0)	40.4 (36.7,44.3)	23.9 (14.3,37.2)	45.7 (43.0,48.3)	0.001
Haiti 2016–2017	37.5 (35.7,39.3)	36.4 (33.0,40.0)	54.4 (42.1,66.2)	37.2 (35.1,39.3)	0.005
India 2015–2016	30.7 (30.4,31.0)	28.7 (27.9,29.5)	32.8 (31.0,34.7)	31.3 (30.9,31.7)	0.001
Kenya 2014	NA	NA	NA	NA	
Malawi 2015–2016	36.1 (34.2,38.1)	29.4 (23.9,35.6)	ND	37.1 (35.0,39.2)	0.020
Mali 2018	56.7 (54.6,58.8)	45.1 (41.2,49.2)	43.1 (31.4,55.5)	59.7 (57.2,62.1)	0.001
Myanmar 2015–2016	26.7 (24.7,28.9)	20.3 (15.5,26.2)	27.4 (19.9,36.4)	27.6 (25.3,30.1)	0.087
Nepal 2016	26.4 (24.0,29.1)	21.5 (18.4,24.9)	24.0 (17.3,32.2)	31.2 (27.5,35.2)	0.001
Nigeria 2018	41.1 (39.7,42.5)	31.6 (29.5,33.9)	48.6 (43.4,53.8)	46.4 (44.5,48.2)	0.001
Pakistan 2017–2018	NA	NA	NA	NA	
Rwanda 2014–2015	15.8 (14.4,17.2)	9.0 (7.0,11.6)	13.4 (5.8,27.8)	16.9 (15.4,18.5)	0.001
Senegal 2017	41.8 (40.2,43.4)	29.7 (27.3,32.2)	48.9 (42.9,54.8)	48.1 (46.2,50.0)	0.001
South Africa 2016	37.0 (32.9,41.3)	41.2 (34.0,48.9)	ND	32.9 (29.1,36.9)	0.108
Tanzania 2015–2016	31.3 (29.6,33.0)	26.0 (23.8,28.3)	38.8 (25.1,54.6)	32.6 (30.6,34.7)	0.001
Uganda 2016	29.1 (27.3,31.1)	24.2 (20.1,28.8)	25.4 (17.9,34.6)	30.2 (28.1,32.4)	0.060
Zambia 2018	29.5 (28.1,30.9)	30.3 (27.8,32.9)	29.9 (24.1,36.5)	29.1 (27.4,30.8)	0.694

Note: The *p*-value is produced from the Chi-square test of the association between the urban poverty cluster variable and moderate to severe anemia. NA means data not available and ND is not displaced because the estimate is based on fewer than 25 observations. Estimates in parentheses are based on 25–50 observations.

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
