# Peer review of "Stunting and Anemia in Children from Urban Poor Environments in 28 Low and Middle-income Countries: A Meta-analysis of Demographic and Health Survey Data"

_nutrients, 2020, doi:10.3390/nu12113539_

Round 1

Reviewer 1 Report

Dear Author, 

your paper "Stunting and Anemia in children from urban poor environments" is interesting and gives to scientific community useful informations, I think the article should be minor revision such as:

1) Data: line 62  you have to define what does it means "till now (present)" because, looking to table 1 it seems that DHS survey have been done in different time in different countries so that you can't say "from 2014 to the present";

2) In table 1 and in the text you have to include the relationship between  population number in different analyzed countries and number of household interviews;

3) Are there some relationships from the previous data? i.e do you have considered an "homohenous ratio" between number of interviews and number of population for countriY?

4) Why don't you have mentioned the data on Iron Deficiency Anemia and environment?

5) As you know anemia "generally speaking" not always is related to "nutrinional deficiency" and probably IDA is a better indicator for nutritional deficiency;

6) Could you please try to insert a new table (table 4) inserting the total data that you have collocted by adding the data of every country in terms of significance?

7) In the Discussion try to be more exact on what your experience teach and on what you can clearly demostrated to the scientific community;

8) Try to insert a "decisional tree" in the discussion (a figure) on what will be the right strategies" that come out from your experience from scientific and economical point of view in order to prevent stunting and Anemia in children from various environments;

9) Try to little improve the english language by checking the text from a native english speaker 

Reviewer 2 Report

The objective of the meta-analysis performed by Assaf and Juan, using data from the DHS surveys, is to describe the burden of two nutritional status indicators (stunting and anemia) in children under 5 in LMIC and to analyze the risks associated with living in urban poor and rural areas compared to urban rich areas. The concept of the study is original. Using the UN Habitat indicator to determine urban poor areas and include it in regressions models is relevant. In addition, the paper is comprehensive and well-structured.

However, an important element missing through the paper is related to differences and changes in population's diet according to levels of urbanization. The “nutrition transition” is a consequence of urbanization and is known to be one of the main drivers of the double-burden of malnutrition worldwide, which is the coexistence of undernutrition (e.g. underweight, stunting, micronutrient deficiencies) and overweight, obesity, and diet-related non-communicable diseases. My major comment is that authors should consider strengthening the introduction and discussion with these concepts.

Please see WHO Policy Brief:

https://www.who.int/nutrition/double-burden-malnutrition/en/#:~:text=The%20double%20burden%20of%20malnutrition,populations%2C%20and%20across%20the%20lifecourse.

Title and abstracts:

  1. In my opinion, the abstract (and if possible, the title) should include the following key words: Demographic and Health Surveys (DHS) and Low- and Middle-Income Countries (LMIC)

Introduction:

  1. The paper would benefit from adding some data about the demographic transition and rapid urbanization worldwide, such as projections of the world’s urban population in 2050
  2. Please consider strengthening the introduction with the concept of double burdenof malnutrition and nutrition transition (see reference provided at the end of this review)
  3. Please check Reference 3, as in the text, it is used to reference trends in anemia.
  4. Please check more up to date publications (and numbers) for references 3, 4 and 6.  

Methods/results

  1. Line 91: Indicates that the regressions are controlled on region. Wouldn’t it be meaningful to add a sub-analysis by WHO regions?
  2. If the authors decide not to run additional analysis by WHO regions, I would suggest ordering the results in the Tables according to WHO regions (Africa, Asia, Latin America…).

Discussion

  1. Line 223. Egypt and Ghana are two countries where rates of overweight and obesity are extremely high. Please consider interpreting the results in light of the nutrition transition, double burden, food environment changes, urban diet … In some places, the urban rich are those with the poorest diet (high calory food, processed food, micronutrient poor food, soda) - For that reason, the assumption that urban rich should have better nutritional status is simplistic.
  2. Line 228: Low access to health services in poor urban areas is only one part of the problem when it comes to nutrition outcomes. Again, diet, food environment, wealth, education are others drivers.

Please find exemples of references that could potentially help the authors.

Dynamics of the double burden of malnutrition and the changing nutrition reality

https://doi.org/10.1016/S0140-6736(19)32497-3

Urbanization, food security and nutrition

https://cgspace.cgiar.org/handle/10568/81593

Food (In)Security in Rapidly Urbanising, Low-Income Contexts

https://www.ncbi.nlm.nih.gov/pmc/articles/PMC5750972/

Estimating the Double Burden of Malnutrition among 595,975 Children in 65 Low- and Middle-Income Countries: A Meta-Analysis of Demographic and Health Surveys

https://www.ncbi.nlm.nih.gov/pmc/articles/PMC6720202/

Child malnutrition in sub-Saharan Africa: A meta-analysis of demographic and health surveys (2006-2016)

https://journals.plos.org/plosone/article?id=10.1371/journal.pone.0177338

Childhood Obesity and Overweight in Ghana: A Systematic Review and Meta-Analysis

https://www.hindawi.com/journals/jnme/2020/1907416/
